# Understanding Social-Ecological Challenges of a Small-Scale Hilsa (*Tenualosa ilisha*) Fishery in Bangladesh

**DOI:** 10.3390/ijerph16234814

**Published:** 2019-11-29

**Authors:** Mohammad Mojibul Hoque Mozumder, Aili Pyhälä, Md. Abdul Wahab, Simo Sarkki, Petra Schneider, Mohammad Mahmudul Islam

**Affiliations:** 1Helsinki Institute of Sustainability Science (HELSUS), Doctoral Programme in Interdisciplinary Environmental Science (DENVI), Faculty of Biological and Environmental Sciences, University of Helsinki, 00014 Helsinki, Finland; 2Development Studies, Faculty of Social Sciences, Department of Geosciences and Geography, Helsinki Institute of Sustainability Science (HELSUS), University of Helsinki, 00014 Helsinki, Finland; aili.pyhala@helsinki.fi; 3WorldFish, Bangladesh and South Asia Office, House 2B, Road 04, Block-B, Banani, Dhaka 1213, Bangladesh; A.Wahab@cgiar.org; 4Cultural Anthropology, University of Oulu, P.O Box 1000, FI-90014 Oulu, Finland; simo.sarkki@oulu.fi; 5Department for Water, Environment, Civil Engineering and Safety, University of Applied Sciences Magdeburg-Stendal, Breitscheidstraße 2, D-39114 Magdeburg, Germany; petra.schneider@h2.de; 6Department of Coastal and Marine Fisheries, Sylhet Agricultural University, Sylhet 3100, Bangladesh; mahmud.cmf@sau.ac.bd

**Keywords:** small-scale fisheries, social-ecological systems (SES), DPSIR, hilsa fishery, impacts, social resilience, sustainable management

## Abstract

Small-scale fisheries (SSFs) have been playing a crucial role in meeting the basic needs of millions of people around the world. Despite this, the sustainability of global fisheries is a growing concern, and the factors enabling or constraining the sustainable management of small-scale fisheries remain poorly understood. Hilsa shad (*Tenualosa ilisha*) is the single most valuable species harvested in Bangladesh waters, serves nutrition, income, and employment to the large population. This study analyzed the state and challenges of hilsa fishery in the Gangetic River systems (Padma and Meghna Rivers) by using two frameworks, namely the social-ecological systems (SES) and drivers-pressure-state-impact-responses (DPSIR) frameworks. Primary data for this analysis were collected by in-depth interviews (n = 130) and focus group discussions (n = 8) with various stakeholders in the hilsa fisheries. The perspectives explored here have been both critical and constructive, including the identification of problems and suggestions for improving the management of this particular social-ecological system. Hilsa fisheries, however, have come under severe threat since 2003 because of population growth, overfishing, pollution, climate change, the disruption of migration routes due to siltation, etc. All these have caused reduced catches and less stable incomes for fishers. This, in turn, has led to poverty, malnutrition, social tensions, stakeholder conflicts, and debt cycles amongst more impoverished fishing communities. These problems have been compounded by improved fishing technology amongst larger-scale ventures, the use of illegal fishing gears, and the non-compliance of government fishery management programs. Recommendations include the promotion of community-supported fisheries, the enhancement of stakeholder’s social resilience, the introduction of co-management approach, an increase in incentives and formal financial supports, and possible community-managed sustainable ecotourism including hilsa fishing-based tourism.

## 1. Introduction

Fishing is one of the primary global industries that supports local livelihoods, food security, and human health, even when carried out at the small scale [1]. Indeed, small-scale fisheries (SSFs) have been recognized as playing a crucial role in meeting the basic needs of millions of people around the world, in both developed and developing countries [2]. Despite this, the sustainability of global fisheries is a growing concern, and the factors that enabling and constraining the responsible management of SSFs remain poorly understood [3]. Many SSFs face growing challenges such as habitat degradation, climate change, a lack of financial sustainability, inadequate equipment and infrastructure, and a lack of access to markets [4]. Additionally, many fisheries suffer from an excess of fishing, which endangers their long-term existence. This is particularly challenging in SSFs settings because controlling access is stimulating [5]. Moreover, the multi-scale management of SSFs remains problematic, and many SSF remain poorly managed [6]. The implementation of proper management practices and the support of local economies are expected to ease the pressure on domestic fish stocks, which in turn is likely to result in an increased food security and an improved quality of life for the local communities [7].

In Bangladesh, coastal resources, including riverine and small-scale marine fisheries, substantially contribute to the national economy, promoting the socioeconomic well-being of impoverished coastal fishing communities [8]. As the single most important fishery, the hilsa catch alone constitutes approximately 12% (0.5 million tons) of the total fish production of 4.134 million metric tons as of 2016–2017 [9]. Fisheries have a total annual value of USD 2 billion, thus accounting for more than 1% of Bangladesh’s GDP and employing approximately 0.5 million fishers and 2.5 million people in the value chain [10]. As an anadromous species, hilsa inhabits a range of habitats and migrates from marine to freshwater for spawning. The hilsa fishery is mainly artisanal and harvested using non-mechanized and mechanized wooden boats. The primary fishing gear includes drift gillnet and set gill nets with different modifications with local names. Many people find employment in hilsa fish marketing chain as fishers, assemblers, processors, traders, intermediaries, transporters and day laborers, including women and children [10,11]. Recently hilsa has been globally recognized as the geographical indicator (GI) product for Bangladesh [12]. This social and cultural significance of hilsa is immense because hilsa is considered the ‘national fish’ of Bangladesh, and is seen as an essential part in religious, social and many festive events [13,14]. To protect the fishery from the overexploitation of juveniles and brood, the government of Bangladesh established six hilsa sanctuaries in the Padma, Meghna, Tetulia, and Andharmanik rivers. In the sanctuaries, a fishing ban has been imposed on catching *jatka* (juvenile hilsa with a size <25 cm) from November 1 to June 30 each year. Another 22-day country-wide closure on fishing brood hilsa is implemented during the full moon in autumn (October month) to ensure safe breeding of hilsa [10,11].

SSFs are increasingly conceptualized as integrated social-ecological complex adaptive systems, or SESs, in part because of the type of problems they exhibit [15,16]. To allow for a more sustainable form of natural resource management, a better understanding of the cause–effect relationships between anthropogenic impacts and socio-environmental components is fundamental [17]. The drivers-pressure-state-impact-responses (DPSIR) framework is a useful tool to identify and describe these links in a meaningful way to resource managers and policymakers [18]. Additionally, the DPSIR approach has been recommended, particularly for the study of fisheries [19]. Despite the extensive use of DPSIR in Europe, the framework is still not well known, nor has it been applied in fisheries sectors of developing countries like Bangladesh. Therefore, this paper puts forth the application of DPSIR and discusses the ways upon which it could be further elaborated, intending to identify responses that can enhance the sustainability of this fishery in Bangladesh. Moreover, given that hilsa is a transboundary species, if good practices could be identified and established in Bangladesh, these could also serve as standards for similar programs in neighboring countries like India and Myanmar.

In this study, DPSIR is used to concretize broader SES dimensions. The SES and DPSIR approaches are used as twin conceptual frameworks to consider the more general objective of this study, namely, how to enhance the sustainability of an SES by developing the application of DPSIR. To examine this, we set out to answer the following research questions:What is the present state of the hilsa fishery?What are the driving forces and pressures in the hilsa fishery?What are the impacts of driving forces, pressures and changes of state on human well-being?What are the possible responses to increase the sustainability of the hilsa fishery?

Below the theoretical frameworks used are firstly presented, followed by a description of the study methods. The results are then presented, followed by a discussion, conclusions, some limitations and further recommendations.

## 2. Theoretical Framework

### 2.1. Social-Ecological Systems (SES)

An SES is an ecological system involvedly linked to and affected by one or more social systems [20]. In other words, in an SES, the role of humans is seen as an essential part of any conservation effort, due to positively enhancing interactions and feedback loops between the ecological and social subsystems [21]. These subsystems include active links related to people’s knowledge (often local or traditional) and management institutions, as well as the rules and norms that mediate how humans interact with the environment [22]. Likewise, SESs are nested, multilevel systems that provide essential services to society, such as the supply of food, fiber, and energy [23]. Ecological endowments, in turn, help shape social and economic systems and occur in a variety of scales, from local to global [24]. Indeed, some scholars have suggested that sustainability science should focus on these linked "social-ecological systems" [25,26]. Additionally, the loss of natural resources across various ecosystems, such as fisheries, forests, and water resources are an increasing concern worldwide [27]. The understanding of the processes that lead to improvements in, or deterioration of, natural resources are limited, as scientific disciplines use different concepts and languages to describe and explain complex SESs [28]. A mounting body of literature supports the idea that small-scale fisheries, both inland and coastal, can be understood as an integrated and critical social-ecological system that incorporate both humans and nature [29,30,31,32].

### 2.2. Drivers-Pressure-State-Impact-Responses (DPSIR)

The Organization of Economic Cooperation and Development (OECD) and the European Environment Agency (EEA) have developed the DPSIR framework as a tool for the analysis of SESs (Figure 1) [33].

According to DPSIR terminology (Figure 1), social and economic developments (driving forces, D) exert pressures (P) on the environment and, consequently, the state (S) of the environment changes. This leads to impacts (I) on ecosystems, human health, and society, which may elicit a societal response (R) that feeds back on driving forces, on state or impacts, via various mitigation, adaptation or remedial actions [34]. Thus, the DPSIR approach has been described as a causal framework for describing the interactions between society and the environment [35]. The DPSIR framework has been applied worldwide across a range of levels and settings, from the global to national scales [36]. After its adoption by the EEA in 1995, DPSIR became popular in studies involving the management of nutrient fluxes in marine environments [37], integrated coastal management [38], development in catchment areas, and offshore wind power generation [39]. DPSIR has also been used for parallel assessments comprising environmental and socio-economic perspectives and impacts [40].

## 3. Materials and Methods

### 3.1. Study Areas

This study was conducted in four hilsa fishing communities in Bangladesh, in the villages of Rahmatpur and Sudirpur (Kalapara Upazila of the Patuakhali district) and Uttar Bagula and Dakxin Bagula (Haimchar Upazilla of the Chandpur district). These areas are hereafter referred to as Study Area 1 (Rahmatpur and Sudirpur) and Study Area 2 (Uttar Bagula and Dakxin Bagula). These areas were chosen because both study areas are situated in the sanctuary area of a hilsa fishery declared in 2005. Additionally, the communities in both study areas have relatively homogeneous populations, where hilsa fishing is their principal occupation. They rely on the hilsa sanctuaries for part of their livelihood, be it in the form of fishing itself or fish trading, fish drying, net mending, boat making, or boat repairing.

Study Area 1 comprises two villages, with a total population of approximately 12,000 people altogether, situated near the Andharmanik River, Kalapara Upazila, the Patuakhali district and in the south-western part of Bangladesh (Figure 2). The Andharmanik River is one of the tidal coastal rivers of the Ganges–Padma system of Bangladesh that flows into the Bay of Bengal through the Patuakhali district. This river is famous for its importance in providing a breeding ground and nursery for hilsa. Hence, the government declared it as the 4th hilsa sanctuary and set a closure period for the juvenile catch (November–January) and for breeding female hilsa (October–November) [41].

Study Area 2 consists of two villages located near the lower Meghna River, Haimchar Upazila, the Chandpur district and in the southeastern part of Bangladesh (Figure 2). Approximately 8000 people live in Study Area 2. The Meghna River is one of the breeding zones for hilsa fish, and the government declared 100 km of the lower Meghna estuary as a hilsa sanctuary in 2005 in order to preserve the juvenile and brood hilsa fish [11].

### 3.2. Methods

In this study, a combination of primary and secondary information sources was used. For empirical data, we utilized a set of qualitative methods, namely in-depth interviews and focus group discussions (FGDs). Qualitative methods were used to answer questions about experience, meaning and perspective—most often from the standpoint of the participant [44,45,46].

Data were collected into two phases. The first phase was conducted from December 2016 to February 2017 in Study Area 1 and the second phase from November 2018 to January 2019 in Study Area 2. The concept of saturation was the core guiding principle to determine sample sizes in this study [47]. Interview respondents were selected using both purposive and snowball sampling strategies [48]. Altogether, 130 in-depth individual interviews were carried out. Of these, 120 were with hilsa fishery stakeholders, and in each of the four villages, interviews were undertaken with 20 individual hilsa fishers—mostly men (n = 15) but also a few women (n = 5). A further ten interviews were conducted with other stakeholders, including fish traders (n = 2), boat owners (n = 2), money lenders (n = 2), local government representatives (n = 2), and local governments administrative personnel’s (n = 2) (see Table 1). Along with hilsa fishery stakeholders, we interviewed academics (n = 4), local NGO (non-government organization) representatives (n = 2), environmental specialists (n = 2) and aquaculture specialists (n = 2). The individual’s interviews lasted some 40–60 minutes on average and were recorded with prior consent.

In addition to the in-depth interviews, we conducted eight FGDs—two in each village. One FGD was carried out with only hilsa fishers (n = 8), including men (n = 5) and women (n = 3), and another with a combination of all stakeholder groups (n = 10) (Table 2). The FGDs lasted on average 60–70 min and were digitally recorded with permission from all participants.

Interviews were semi-structured but allowed for open-ended conversation. Where informants preferred not to be recorded, extensive notes were taken during the interview. The questions were adapted based on the role and representation of the interviewee (see Appendix A- Semi-structured questionnaires for in-depth interviews/Focus group discussions). A list of topics and possible questions for each interviewee focused on the following overarching themes based on the research questions of the study: (i) the state change of the hilsa fishery, (ii) driving forces and pressures in the hilsa fishery, (iii) the impacts of driving forces, pressures and state changes in terms of hilsa fishery stakeholders well-being, and (iv) possible responses to increase the sustainability of the hilsa fishery. Interview questions revolved around the following areas: perceived reasons behind the decline of hilsa catch, the participant’s perceptions and attitudes towards the strengths and weaknesses of the current hilsa conservation and management policies, including the efficacy and effectiveness of incentives (e.g., for not catching fish during ban periods), and questions on fairness and equitability vis-à-vis distribution; In the FGDs, a list of similar questions was developed beforehand, but we allowed for new questions to emerge during the discussion.

In addition to the above, secondary data were collected from daily newspapers and study reports, by NGOs and local universities working with the small-scale fisheries (mainly the hilsa fishery) in the coastal areas of Bangladesh, and with associated legislation issued by the government of Bangladesh. Moreover, was reviewed the scientific literature related to hilsa fishery management in Bangladesh (from 2001 to 2019) (see Appendix A). This secondary data was particularly useful in providing information about illegal and unlawful activities taking place in the hilsa fishery. Additionally, these documents contained information relevant to the specific case studies, including the types and characteristics of natural resources, socio-economic characteristics, management institutions, and governance systems. The reviewed literature thus helped to better contextualize the interviews and observation data [49], and they proved useful in validating such data [50].

The qualitative data from the in-depth interviews and FGDs were transcribed, translated from Bengali to English, and analyzed using thematic analysis for related themes [51]. Thematic analysis is particularly useful for drawing classifications and contemporary themes (patterns) that relate to the data. This illustrates the data in detail and deals with diverse subjects via interpretations [52]. Themes were related to specific research questions and guided further data analysis. Additionally, direct quotations were used to support and clarify the perceptions of the respondents. Initial analyses of the data were made jointly with the respondents in the field to eliminate personal biases in interpretation.

## 4. Results

The results suggest that the main driving forces for less catching by fishers are the use of illegal fishing gear, overpopulation in the coastal areas, overfishing, the harvesting of juveniles, river water pollution, climate change, dam constructions in the upstream, and the cross-border smuggling of hilsa. These have led both to compromise with management strategies as well as disincentives and further pressures on the hilsa fishery. These pressures include habitat destruction and biodiversity loss, which in turn result in less hilsa catching, poverty, malnutrition, stakeholder conflicts, insecurity and social tensions. To address these challenges, multi-level responses are recommended for the sustainability of the hilsa fishery, including enhanced social resilience of the fishing community, increased incentives for all fishers and major stakeholders at the ground level, promoted community-supported fisheries, improved financing mechanism for the fishers, and introduced hilsa fish-based eco-tourism.

### 4.1. Drivers and Pressures

Population growth and increased number of fishers and non-fishers: The population of Bangladesh is increasing day by day. In the year 2003, the total population of Bangladesh was 132 million, and in the year 2019 it has been recorded as about 168 million. An elderly respondent in Study Area 1 cited population increase as causing low hilsa fish catches because too many fishers are competing for too few fish. Thus, fishers have turned to illegal means to catch enough fish even for meeting just their own subsistence needs. The over-crowded situation in the hilsa fishery was further explained by a 50-year-old fisher in Study Area 2 as follows:

“*During my youth, I rarely saw any other fisher on the water within a kilometer of me. Now, nets and other fishing equipment are set as close as the fingers on my hand. Thus, there is fierce competition for fishing space, which can get nasty.*”

Due to free access to water bodies and the progressively rising price of hilsa, many part-time fishers also get involved in fishing to make extra money. Therefore, the number of fishing boats is increasing, which is increasing the intensity of fishing. Nowadays, fishers from other districts or areas come to catch hilsa in the localities we studied. Not all of these have hilsa fishing as their primary occupation, and they thus do not care about ban periods or restrictions on catching juveniles and broodstock. Fishers told stories of these sorts of operators in both studied areas during the interviews and FGDs.

Overfishing: One academic and expert informant who is involved with hilsa fishery research stated during his interview that the *Jatka* (juveniles) and matured hilsa fish are being caught indiscriminately during their downstream migration from the Meghna River to seawater in the year-round cycle. Such intense localized fishing pressure could inevitably reduce the adult and breeding fish stock, with a possibility of overfishing of hilsa in coastal and marine waters of Bangladesh. He further added that, during the spawning migration (from marine water to freshwater and vice versa), vast numbers of gravid and immature fish are caught using a variety of different fishing gear. According to some informants, the hilsa fishery in Bangladesh suffered from a combination of factors such as the indiscriminate harvest of gravid fish and the indiscriminate catching of *jatka*.

Juveniles and broodstock harvestings: To allow for uninterrupted spawning, any catch of brood hilsa is banned in all the primary breeding grounds for 22 days around the full moon in October, the Bengali month of *Ashwin*. However, fishers tend to violate the rules as the fish is available in abundance and is vulnerable during breeding seasons. One hilsa fisher in Study Area 2 stated: “*The highest numbers of brood and ripe hilsa are caught every year during the breeding season, because the brood hilsa has high market demand locally and internationally.*”

Water pollution and climate change: In Study Area 1, during the FGD sessions, hilsa fishers stated their perception about climate change and its impacts on hilsa fishery as follows: “*We can assume that the climate is changing. Temperature is increasing, so salinity is increasing in the estuary. It is not suitable for the hilsa and could act as a hindrance to hilsa migration. The water quality of the river is degrading, siltation is increasing, and there hasn’t been enough rain during the rainy season for brood hilsa (broodstock) to lay eggs.*”

Dams and barrier constructions: Due to the construction of different flood control drainage and irrigation projects and barrages, water flow in Bangladesh has considerably reduced, and hilsa fisheries have been severely affected. Additionally, the establishment of the Farakkah barrage on the Ganges–Padma river system, located 18 km upstream of the Bangladesh border, has led to reduced water flow and increased siltation in the upper courses, with significant implications for the migratory pattern of this anadromous fish. This perspective was added regarding the effects of dam and barrier construction on hilsa fisheries during an interview with a government official and fisheries officer from the Department of Fisheries at Chandpur (Study Area 2).

Changes of migratory routes: An academic and hilsa researcher opined that the migratory routes and spawning grounds of hilsa were disturbed, displaced or even destroyed because of various anthropogenic activities, climate change effects, increased siltation and the rising of the river basins. Additionally, it has been found that the significant spawning areas have been shifted to the lower estuarine regions of Hatia, Sandwip and Bhola in the southern part of Bangladesh. This analysis was supported by an experienced fisher from Study Area 2: “*Sedimentation has increased, and many sandbars have been formed in the riverbeds of Bangladesh, including the Meghna River. I think these sandbars caused blockage in the migratory route of hilsa and reduced the nursery areas for the fish fry.*”

Smuggling of hilsa: A local hilsa fish trader in Study Area 2 stated that due to the high transportation costs from the catch point to the main Bangladeshi markets, as well as greed for a higher price, large, high-quality hilsa are often smuggled to India, pushing the price up in the local market. Smugglers can also turn a considerable profit selling hilsa on the Indian market. During the FGD, another fish trader described the hilsa smuggling situation as follows: “*India likes to import hilsa through legal channels, but due to temporary bans on exporting hilsa, there is a tendency towards smuggling, as well as to avoid complicated customs checks.*” Considering the continued short supply of hilsa in the markets and its high price, the Bangladesh government banned the export of hilsa on August 1, 2012.

Improved fishing technology: During the FGDs in Study Area 1, the respondents stated that improved fishing technology is part of the problem: “*Modern and improved fishing technology is one of the reasons for hilsa depletions. Earlier, we used normal cotton nets to catch hilsa, and the mesh size was big. Hence, juvenile hilsa could easily pass through the net. The use of a modern jal (a fishing gillnet made of monofilament synthetic nylon fiber) with a small mesh size began in the early 1990s. After that, we we able to catch more hilsa in a short time. Also, some fishers are using motorized boats rather than rowboats nowadays, as we get a loan from the moneylenders to build such boats. Sometimes, we use more powerful engines in our boats than the law enforcement authorities have, so we can easily get away with any illegal gear or (juveniles and broodstock) catch we happen to have.*”

Use of illegal fishing gear: In the hilsa fisheries, using gillnets with a mesh size of less than 10 cm is prohibited by law. Different types of fishing gear are used in hilsa fishing including drift gill nets, fixed gill nets, seine nets, *current jal* (monofilament synthetic nylon fiber), the *behundi jal* (set bag net) and *charghera* nets (a type of surrounding net). Among them, according to fishers in both study areas, small meshed current nets, *behundi* and *charghera* nets were identified as the most destructive, used mainly to capture juvenile hilsa (*jatka*). Additionally, most of the respondents stated that the extensive use of set bag nets was one of the leading causes of hilsa depletion. One elderly fisher from Study Area 1 explained why fishers tend to use illegal fishing gear: “*If we use big mesh size net to catch hilsa, we get only a few. It is tough to survive by selling such a small amount of catch, as we must pay for the boat, fishing gear and wages for our hired help. Also, we took a loan from local moneylenders in our crisis time. Considering all the issues, including the huge pressures from the moneylenders to pay the loan, we use the small mesh size net and can also catch the juveniles. We know it is illegal, but we must survive.*”

Imposed ban period: To increase the size and sustainability of the hilsa catch, the government of Bangladesh has imposed fishing ban periods in six hilsa sanctuary areas. The incentive-based hilsa conservation program has four main activities: awareness-raising, providing food to fishers’ households, strictly enforcing ban periods, and supporting alternative income-generating activities. However, these ban periods have had a significant impact on the livelihoods of hilsa fishers in the study areas, causing desperation which in turn has put immense pressures on hilsa stocks. This was evident from the fisher’s responses in the FGDs conducted in Study Area 2: *“The ban period, popularly known as the ‘obhijan’ (expedition), is one of the biggest shocks in our life, and it has placed us under severe financial duress. To overcome the situation, we sell off family properties, participate in seasonal migrations, reduce our daily food intake, compel other family members to get jobs, take out loans from money lenders and NGOs at a very high interest rates, and, finally, we get involved in illegal fishing, including for juveniles and broodstock during the ban period; even though we realize that the ban period is good for us in the long run.*”

Inadequate allocation of incentives by the government: The government of Bangladesh introduced an incentive program of 40 kg of rice per month per hilsa fishing family during the ban period. However, fishers complained about the distribution of incentives as such during the FGDs in Study Area 1: “*The incentives rarely come in time to feed our families during the fishing ban period. Some fishing households have seven-to-eight members or more. Do you think 40 kg of rice is enough for those households for one month? Sometimes, we get only 30–35 kg of rice instead of 40 kg, if we ask why we are getting only 35 kg of rice, the officials answer that it is due to handling costs or their limited transportation budget or the like.*”

Similarly, not all fishers were included in the beneficiary lists. Fishers claimed mismanagement and official corruption among local government administration, the chairman of the Union Parishad (smallest administrative unit), and bureaucrats who prepare the beneficiary lists: some deserving fishers are excluded from these lists, and some non-fishers are given benefits intended as incentives not to fish during the ban periods through favoritism. Such competition for inclusion in the government compensation scheme, together with irregularities in its distribution, has sometimes caused a spike in tension. During the combined FGDs in Study Area 2, there was a disagreement about the issues. However, the local government administrations personnel accepted a maximum of the issues raised during the discussions.

Improper fishing regulations and compliances: The use of a *current jal* with a mesh size of less than 4.5 cm was banned in 1988. The 2002 amendment to the 1950 Fish Act states that ‘no person shall manufacture, fabricate, import, market, store, carry, transport, own, possess or use a *current* net.’ The penalty for violating this law is imprisonment and a fine. Fishers from Study Area 2 stated during the FGDs: “*We know some rules and regulations exist in hilsa fishing. As an example, current jal is illegal for catching hilsa, and to buy a current jal, we have to pay much money. When the law enforcement party catches us during the ban period, they seize or burn the net. Afterwards, we repurchase the nets from the market by taking a loan from the local moneylenders. To pay the loan, we go illegal fishing. Also, law enforcement personnel do not seize the net from the market but seized it from us. If law and enforcement personals take steps to seize all the illegal nets from the market, we cannot buy it from the market.*”

A short summary of drivers and pressures in the hilsa fishery stating their problems, category, effects, possible solutions and alternatives were presented in Table 3. The findings were summarized as approved by the fishers and other stakeholders during the FGDs and ranked these according to the importance given by interviewees and FGD participants.

### 4.2. States

Biodiversity loss: Different types of equipment are used by the hilsa fishers to catch juveniles and mature hilsa. Changes in the biodiversity of the Meghna River were described by one fisher in Study Area 2 as follows:

“*Twenty years ago, I saw different types of fish in this river, and fish diversity was quite high. as it served as a nursery, breeding and feeding grounds for many fish species. After the introduction of current jal [small mesh monofilament fixed gill net]—an extremely effective way to overexploit juveniles—I hardly see many of species that I used to see or find in this river. In my opinion, current jal is one of the most harmful nets and is responsible for the decline in fish populations in the Meghna River.*”

A fisheries expert and NGO official in Study Area 1 expressed his views about the biodiversity loss thusly: “*The local fishers extensively use set bag nets in the Andharmanik River. It is used to catch fishes like Chital (Chitala chitala), Ilish (Tenualosa ilisha), Bele (Glossogobius giuris), Taposi (Polynemus paradiseus), Poa (Otolithoides pama) and Golda chingri (Macrobrachium rosenbergii). All these species are caught at the juvenile stage. These undersized fish are locally known as ‘guramas,’ and every kilogram of this guramas contains thousands of larvae of commercially valuable fish. The significant portion of these commercially valuable species are unable to grow in a mature stage due to set bag net fishery, and the result is a failure in recruitment and a decline in total production.*” He further spoke of uses of set bag nets that do not allow even the fingerling larvae to escape. These are spread across the river, and thus they are the leading cause of hilsa stock depletion.

Habitat destruction: Additionally, known as habitat loss, this occurs when the natural shelters that are home to different species are destroyed. The views of hilsa fishers regarding the destruction of hilsa habitats were evident during the FGDs in Study Area 1: “*The Andharmanik River is the home of hilsa. I have been fishing in the Andharmanik River for the past 30 years, and the water was clear enough. Not polluted as it is now. The water color has changed, and I can smell a bad odor in the water. From my understanding, it is happening as lots of industries have been built on the riverside, and the industries have thrown by-products full of toxic chemicals into the river water. Certainly, these chemicals are not good for the fish, and their food (plankton) and the fish are migrating other areas to survive.*”

The 40 km stretch of the Andharmanik serves as a significant corridor for hilsa migration from the sea. The government has approved many coal-fired powers projects near the hilsa sanctuaries in the Patuakhali coastal area. Due to this, many vessels carrying coal, possible oil spills, fly ash, and the discharge of water used by the plants could seriously harm the plankton-rich waters where hilsa breed. These power plants significantly disturb the breeding and nurturing of hilsa regardless of the safety measures they take concerning the environment. An environmental expert expressed his views about the possible consequences of habitat destruction in Study Area 1, as stated above. Further, he added, the Andharmanik River in Patuakhali is known as the hilsa hub. Thousands of people involved in this trade could lose livelihood if fish production drops due to water pollution caused by the power plants.

### 4.3. Impacts

Fewer hilsa catches: During the interviews and FGDs in both studies, hilsa fishers’ areas stated that they are getting fewer catches nowadays, mostly because of population growth, overfishing, climate change and the illegal catching of juveniles and broodstock. One fisher from Study Area 1 pointed out, “*Hilsa is disappearing from the Andharmanik River. I have been fishing for the last 30 years. If I compare today with 20 years ago, I can see the differences. It took only a few hours fishing, and our boat deck became full of hilsas. Today, with three persons in my boat, after eight hours, only half of our deck is full. If we sell this fish to local markets, the dadondar will take his interest on the money that we borrowed. Afterwards, we hardly have money to spend on ourselves and our family. We are going through hard times and must change our profession.*”

Poverty: “*Well, you know, we are the full-time hilsa fishers and imposition of a hilsa fishing ban brings economic hardship to us, and we do not have other alternative occupations. The compensation (not to fish during the ban) that we receive from the government is of insufficient quantity, requiring extra cash support for satisfying other essential costs for the family such as children’s education. Thus, the ban on hilsa fishing pushes us into poverty. Also, other forms of punishment such as seizing hilsa catch, monetary fines and imprisonment make us vulnerable to economic crises.*”—fishers in the FGDs, Study Area 2

Malnutrition: During the FGDs, most of the respondents raised concerns about year-round food insecurity. They reported food insecurity, particularly during the banned fishing period in the sanctuary. Fishers in Study Area 2 during the FGDs agreed about the malnutrition risk in these terms: “*The hilsa fishers go through food insecurity for four-to-five months of every year, and the reason for this food insecurity is an off-season of fish catches. Although the bans have resulted in a higher total hilsa fish catch, the two-month complete bans in five sanctuaries have created a range of negative impacts on the dependent livelihoods. During the non-fishing season and banning season, the condition becomes more pitiable. We cannot cover the expense of even our basic food. Only a few fishers—those who had agricultural land—had food security for the whole year. Earlier hilsa was widely accessible to consumers in all income groups. Still, due to a decline in the wild catch, its price has increased several-fold, making it impossible for poorer consumers like us to afford it. We know that hilsa is rich in nutrients, but we cannot eat it to fulfil our nutritional requirements. Children and pregnant women suffer most from lack of nutrition during this time.*”

Conflicts: Thousands of people are involved in hilsa fishing as well as different forward and backward linkage activities in the fish chain (fishers, aratdar/mahajan/dadondars-money lenders, local government administrators, NGOs, departments of fisheries, and law enforcing agencies including police and coast guard). Participants summarized the stakeholders’ conflicts in the hilsa sanctuaries by saying that sanctuaries have increased conflicts among them (competition for inclusion in incentive schemes and competition for fishing space) and negatively impacted their income. Additionally, sanctuaries have resulted in their household food consumption becoming insecure, and sanctuaries have only benefitted the coastal ecosystem. Conflicts between fishers and money lenders were summarized in Study Area 1 as follows: “*We the hilsa fishers get dadon (loan/micro-credit) from dadondars or local NGOs to buy and maintain our fishing equipment, to buy daily necessities, and to get protective security services. But such debt bondage makes us sell fish at a lower price; sometimes we sell fish to other buyers to get more profit or delays to pay the loan. In such case of default, dadondars/NGOs personnel seize our all assets, including our boats, fishing gear and the farmland.*” Fishers also described their conflicts with other stakeholders thusly: “*Department of Fisheries personnel do not consider our opinions in developing fisheries management strategy, and law enforcing agencies (police, coast guard) accuse us of illegal fishing, bribery and harassment.*”

Debt trap: In the Study Area 1 FGDs, hilsa fishers stated how they remained in a debt cycle all the year round: “*Hilsa fishing is very capital-intensive, and our income from fishing is impoverished. To buy fishing boats and other equipment, we take a loan from local individual money lenders (dadondars), as we do not have access to the formal credit markets such as banks (government and non-government). Banks do not allow us to take loans because we do not have land property. As a result, we remain dependent on informal credit mechanisms, like the dadon system ( an unwritten contract between the fisher and the money lender, whereby the money lender requires that the fisher sells the fish to him, or money lenders gets a specific commission when fish is sold to a third person). Thus, the dadon system binds us to the money lender in a debt cycle as we cannot pay the loan fully.*”

During one FGD, respondent hilsa fishers in Study Area 2 added further: “*Sometimes we are caught by the law and enforcement authority, if we fail to pay demanded money, they sent us to jail and burned our fishing nets. To free us from them or to save our nets or to buy new fishing nets, beside the local dadondars, we used to get a loan from the local micro-credit organizations with a high-interest rate. Though sometimes we had a good catch, after meeting all the expenses, we had very little to live on. Excessive pressure to pay loans or cover the instalments sets by the micro-credit organization, we take the risk to engage in illegal fishing in sanctuaries during the ban period.*”

Destitution of fishing security: Hilsa fishers are in fear of increasing pirate attacks during the peak season, and the fishers discussed this in the FGDs. One fisher from Study Area 1 shared his experiences as follows: *“One day we (four fishers) went hilsa fishing in the Andharmanik River. Suddenly, we were captured by pirates. We were given food only one time in a day and little water to drink. They took our boat and fishing nets. Also, they demanded ransom from our family to let us go. Later, our family took a loan from the dadondars and ransomed us from the pirates. It was a painful moment that cannot be described. I urge to ensure security for us while we are fishing.*”

Reduced social bonding and heighten social tensions: Hilsa fishery stakeholders stated that overall social bonding is reduced, and social tensions are increased in the area for several reasons. One fisher from Study Area 1 described the situation thusly: “*Nowadays, there is strong competition for fishing space in the Andharmanik sanctuary that often leads to conflicts when fishers try to spread nets close to one another. In turn, such a situation, leads to a loss of property or even physical harm, often spilling over into other communities, further increasing social tensions on land.*” Another fisher (non-mechanized boat owner) from Study Area 2 said, “*There are conflicts between mechanized and non-mechanized boat owners. Fishers with non-mechanized boats (rowboats, without an engine) and those with mechanized boats blame each other for illegal fishing, though both types of fishers continue fishing during the banned period. See, I have limitations, including a small boat and no engine, I can only harvest a smaller catch of illegal fish, and mechanized boat owners often catch a huge amount of fish. Also, we are often caught red-handed during raids by law enforcement officials (police and coast guard). Mechanized fishers, on the other hand, escape more easily. Sorry to say, large mechanized boats are owned by the local elite, who typically give bribes to the law enforcement officials and can continue fishing at night during the banned seasons. Also, if there is any raid, they generally find out about it in advance from their sources in the police station.*”

### 4.4. Responses

Hilsa production has increased over the years; however, there are widespread concerns about fishers’ socioeconomic conditions due to exploitation in the complex value chain and lost earnings during fishing bans. To compensate for the economic loss, the government provides financial incentives in the form of rice and some support for alternative income-generating activities. Reportedly, these interventions have achieved limited success. The participants of this study during the in-depth interviews and FGDs stated some possible responses that should be considered for the sustainability of the hilsa fishery. These were:

Proper compensation scheme: During the FGDs, Hilsa fishers in Study Area 1 summarized the importance of the adequate amount of incentives. They mentioned: *“The incentives should be delivered before the ban period starts. Only the local hilsa fishers should be included in the beneficiary list. Instead of local government administration. The Upazila fishery office can distribute the incentives among fishers.*” Also, a fisher from Study Area 2 urged: *“Forty kilograms of rice/month is not sufficient for the family. We need 90 kg of rice and BDT. 1,000 (Euro 12) in cash per month during the ban period to meet essential expenses.*”

Diversified income-generating activities: Participants in the FGDs in Study Area 1 said: “*To compensate for the loss of income during the banned periods, we need support for alternative livelihood options, including our women. Some examples of livelihood options are nets making, cage culture/fish farming, poultry rearing (ducks and chickens), small dairy projects (keeping goats or cows), starting-up a new business (small grocery shop/tea stall), plant nurseries, gardening, and handicrafts (doll making)*.” Another fisher in Study Area 2 added: “*We will not go fishing during the ban period if the government takes initiatives to teach us how to do farming and gives us goats/cows/chickens. We would then engage in farming activities*”.

Use of fishers’ local-ecological knowledge: In the interviews and FGDs, it was apparent that elderly hilsa fishers were excellent sources of local ecological knowledge about the fishery. An old fisher in Study Area 2 stated, “*I have been fishing almost 30 years in the Haimchar areas in the Meghna River. I know what types of food hilsa eats and where the feeding grounds are, where the breeding grounds are, what types of water color are favorable for the hilsa broodstock to lay eggs in and when the suitable times are for them to lay eggs. Also, I am aware of seasonal fluctuations in fish stocks, the seasonal movement of the fish, and other environmental factors, including temperature and salinity effects on hilsa migration.*” Another fisher from Study Area 1 added: “*Although we are illiterate, we gained fishing experience by fishing in the Andharmanik River for a long time. I think that local fisheries department officials should discuss hilsa matters with us and use our local knowledge in setting the dates for fishing bans and the geographical boundaries for fish sanctuaries and in formulating other fishery policies”.*

Fisher-friendly credit systems: Though fishers can get loans with high interest or specific agreement from the local moneylenders or NGOs working with micro-credit issues; they would like to get out of the debt cycle. During the FGDs in Study Area 1, fishers said, “*If we can sell the hilsa to the different buyers then we can get a better price and save money for buying the boats, fishing gear and household necessities. Indeed, the government can introduce a fisher-friendly credit system through the local banks so that we can get a loan with easy conditions”.* An academic in Study Area 2 expressed his views about the fisher-friendly credit systems thusly: “*The formal credit system should be reformed to meet the needs of hilsa fishers, considering the uncertainty of the fish catches, the lack of collateral and co-signers, and the need for flexibility. Pioneering credit systems should be introduced to enhance hilsa fishers’ adaptive capacity and to diversify the local economy without leading to increases in fishing pressure on already over-exploited resources. Access to credit that enables fishers to invest in other sectors such as farm agriculture, aquaculture, or poultry/goat rearing would reduce pressure on fish stocks while providing for livelihood diversification.*”

Strict compliances and enforcements: There are several laws regarding hilsa conservation. However, there are limitations in their implementation and enforcement. Hilsa fishers summarized the situations during the FGDs in Study Area 1 thusly: “*Well, we know some laws but not all. The government should take steps to make us aware of those rules and regulations through mass media. The ban period should be strictly managed in all the hilsa fishing districts. Current net or monofilament gill net should be banned for all, as they are the main reasons for hilsa stock depletion. We think the punishment should be increased for the manufacturing, fabrication, import, marketing, storing, carrying, transport, owning, possession or use of a current net. There should be laws and punishment if the local government representative fails to distribute the incentives fairly”.* Additionally, an elderly fisher in Study Area 2 said, “*There are lots of small canals that feed into the Meghna River. All the boats are kept in the canals before fishers go for fishing in the river. During the ban period, if law and enforcement authorities close the entrances to the canals that feed into the Meghna River, no boat can go fishing in the river. This will help the patrolling effective and minimize the cost of the government.*”

To ensure the security of hilsa fishers against pirate attacks during the peak hilsa harvesting seasons, during the FGDs, the fishers in Study Area 2 urged: “*The law-enforcing administration should take necessary actions against the pirates. In such a case, coast guards could play an active role because they have modern speed boats and necessary weapons to rescue us (the victims). Also, nowadays we all have mobiles, and mobile networking in Bangladesh is much stronger than it used to be. It is also spread through remote areas. Hence, law-enforcing agencies could utilize such technology by providing a special helpline service to seek help in an emergency while the fishers are out for fishing. This will ensure the rapid movement of the police or coast guard to the place of the pirate incident to rescue us (the victims) and ensure the imprisonment of the criminals”.*

Transboundary initiatives: Hilsa is a major transboundary migratory fish species of Bangladesh, India and Myanmar, as 90% of the world’s hilsa catch comes from these three countries. An academic and a hilsa researcher in Study Area 2 said, “*At present, it is badly needed for Bangladesh, Myanmar and India to work together towards sustainable hilsa fishery management. In Bangladesh, the ban on fishing is strictly enforced, but in India and Myanmar, it is not enforced. A common regulation (e.g., mesh size and bans) should be formulated by Bangladesh, India and Myanmar to restrict hilsa fishing during breeding and juvenile nursing and to promote common management measures for better conservation and enhanced production. Transboundary research on hilsa biology and fishery management can be a good initiative for the welfare of the fishing communities, various hilsa stakeholders, and the sustainability of hilsa fisheries”.*

*Awareness, empowerment and participation:* An Upazila fishery officer, during his interviews in Study Area 1, stated the importance of stakeholders’ engagement and outreach in hilsa conservation. He said: “*Hilsa sanctuaries are in densely populated basins. Resident fishers can voluntarily patrol these areas. If fishers and other stakeholders in the value chain, such as fish traders and money lenders, shoulder some responsibilities for managing these sanctuaries, they are less likely to encourage the violation of the conservation regulations”.* In Study Area 2, during the FGDs, moneylenders (*dadondar*) said: “*We invest huge amounts of money in the hilsa fishery as loans to fishers without any paper works or any security. If we do not get hilsa fish from the fishers, we cannot make a profit (ban period). We know that without the hilsa, we cannot survive. Hence, we also want to conserve hilsa for our betterment. However, the government should think of us and include us in the compensation schemes or other suitable alternatives.*”

## 5. Discussion

### 5.1. Elaborating Policy Responses to Enhance Sustainability in and around Hilsa Fisheries

The problems in an SES can be addressed at a social, economic or ecological level, as well as at a regulatory level through governance and management structures. This requires a societal response from the interested stakeholders of the SES as a means of the capabilities of society to respond to deteriorating situations and on the actions that are undertaken to solve or mitigate damage [53,54]. This is also true for the hilsa fishery, and to make a hilsa fishery SES sustainable, societal responses are necessary. As seen above, different responses were stated by the participants, also shown in the proposed DPSIR framework (Figure 3), including proper compensation schemes for the stakeholders, strict compliances, alternative income-generating activities, fisher-friendly credit systems, the use of fisher’s local knowledge, the awareness and participation of stakeholders, and transboundary initiatives.

In this study, we considered different responses to describe as a remedy to the harmful effects of impacts in the hilsa fishery. Additionally, responses had to be environmentally sustainable (nature-friendly), economically viable (at a reasonable and a supportable cost), socially desirable (wanted by hilsa fishery stakeholders) and administratively achievable (carried out by different ministries including fishery, agencies, NGOs and governments). Finally, potential responses had to make the hilsa fishery sustainable, and the present study propose the following responses based on the frequency and priority given to each by informants in both interviews, FGDs and authors observations.

Compensation-based schemes: These schemes, in which natural resource users are compensated or rewarded to change their destructive and unsustainable fishing practices, have been increasingly acknowledged as an alternative to failed regulatory mechanisms [55]. However, their application in fisheries, where resources (fish) are more mobile and harder to monitor and where property rights are often ill-defined or insecure, remains embryonic [56]. If well designed, however, such schemes could play a significant role in incentivizing fisher or coastal communities to conserve, restore and sustainably manage their resources [57]. A growing number of examples from around the world point to ways in which adding incentives to existing ‘regulatory’ schemes can make them more effective in protecting both environments and livelihoods [58].

The incentive-based hilsa conservation program has four main activities: awareness-raising, providing food to fisher’s household, a strictly following of the ban period, and support for alternative income-generating activities [59]. However, the present study found that the scheme is not without its flaws, and knowledge gaps highlight the need for further research into the effects the sanctuaries are having on hilsa stocks and how the system is reaching and affecting those people who depend on the fish for a living, particularly the poorest and most marginalized fishing communities. Additionally, as found in the present study, the compensation distribution is currently not considered enough, adequate, fair, or equitable according to the local fishers. Instead, small-scale fishers remain disproportionately and negatively affected, at least in the short term, by limited harvests. Yet, they continue to have relied heavily on fishing as their primary livelihood activity [60]. Hence, schemes need to make a much more careful assessment and prioritization according to needs and access.

Enhanced compliance with regulations: Regulatory non-compliance and a lack of capacity for carrying out enforcement operations are significant barriers to the effective management of the hilsa fishery [59]. Our findings demonstrate that economic incentives are the most important single factor that influence compliance behavior in hilsa fisheries. Additionally, our ideas support the observations that most fishers have an opportunistic approach to non-compliance and consider non-compliance behavior in situations where there is a significant economic gain to be obtained, including an assessment of the risk of detection and following (primarily economic) sanctions [61].

Promoting local ecological knowledge (LEK): Increasingly, local ecological knowledge (LEK) is recognized as a valuable tool for understanding social-ecological change and the adaptation strategies designed and implemented by local populations [62]. The present study highlights that hilsa fishery stakeholders, including the fishers, are knowledgeable in ecological processes, and, more importantly, they are familiar with hilsa feeding behavior, spawning time and areas, and water quality, knowledge which could be used in hilsa fishery management as well as providing valuable information for the design of new sanctuary areas.

Enhancing the social resilience of the fishing community: The resilience of both the fishing community (as a social unit) and individual community members are tightly linked to the resilience of the overall fishery SES [63]. The viability of the hilsa restoration project is at risk through over-exploitation, non-compliance with regulations, and conflicts over resource use. To avoid such a situation and to sustain the natural resilience of the hilsa fishery, it is essential to enhance social resilience—the ability of individuals and communities to cope with disturbances and their means of adapting, transforming and potentially becoming stronger in the face of socio-economic, political and environmental challenges [12,64]. For strengthening social resilience at the community level, the authors suggest specific measures including building community networks, developing community infrastructures, updating existing rules and regulations, providing alternative means of generating income for fishers during the crisis periods (e.g., fishing ban periods), and the more active sharing of responsibility between stakeholders and government for management of the hilsa fishery.

Incentives for all stakeholders: The failures of traditional management suggest not only that total harvests must be set appropriately but also that the most significant predator, (i.e., fishers), be provided with the incentives to fish sustainably [65]. However, not only the fishers but also other stakeholders such as money lenders—who are also impacted by the hilsa fishing ban—should be included in the compensation schemes. That said, increasing the number of beneficiaries of the compensation scheme is heavily dependent on mobilizing additional funds [66]. The authors suggest a mechanism for increasing access to funds which involves reducing the transaction and administration costs of the compensation scheme. Further reductions may be achieved by simplifying the beneficiary selection process, thereby reducing staff salaries, and locally sourcing rice, consequently reducing distribution costs. One way of doing this could be to design financially and logistically plausible predetermined compensation packages and to consult recipient communities and households before decisions are made.

Community-supported fisheries: Introduction of a community-supported fisheries(an alternative business model for selling fresh, locally sourced fish), could be another strategy to avoid higher dependence on moneylenders, like that of community-supported agriculture, in which consumers provide up-front financing in the form of a membership fee and, in return, receive a portion of the harvest [67]. Such initiatives could also be introduced in the Bangladeshi hilsa fishery management scheme to enable more impoverished hilsa fishers to buy boats and equipment in advance and therefore get a fairer market price for their catch.

Development of micro-enterprise: The development of micro-enterprises is a vital strategy to augment the income of small fishers, alleviate poverty, and simultaneously reduce fishing pressure in the coastal area [68]. The development of micro-enterprise strategies for the welfare of the fishers is popular in small scale fisheries of India and Philippines [69]. Besides the economic benefits for the fishers by selling hilsa fish, the hilsa fishery inspires the development of related microenterprises: boat making, net mending, packing, processing, icing, transporting, and the trading and marketing of fish. However, special attention is needed for the strengthening of the linkages between the various types of enterprises in clusters or chains. Additionally, there should be a basic set of training program suitable for microenterprise development, consisting of but not limited to the following: values orientation, organizational strengthening, product development, and marketing. The members of a fisherfolk association must undergo this training program before starting the micro-enterprise to ensure its success [70].

Improvement of financing mechanisms: The financial activities of the fishing villages mainly depends upon the availability of credit at a reasonable cost to enhance production and income [71]. Fishing communities rely on institutional and informal credit sources. Institutional sources are mainly banking/cooperatives, and the friendly credit system includes professional moneylenders (who do not ask what the loan is meant for) and fish traders (who lend money to secure fish supplies) [72].

In Bangladesh, fishers, traders and intermediaries do not have easy access to the bank, and non-government organization credits due to too much official paperwork and collateral arrangements. Hence, well-thought-out microcredit should help to gradually liberate hilsa fishers to escape from a cyclical debt trap and prevent the interest rates they pay from rising when the fishery is closed due to ban periods. In this case, Self-help groups (SHGs) may play a vital role in reducing the vicious circle of indebtedness among hilsa fishers. Microfinancing through SHGs has been recognized internationally as a modern tool to combat poverty and rural development [73]. By expanding activities, these SHGs are now well-set institutions for microcredit systems for different purposes of credit needs of rural communities in India [74,75]. However, there is an urgent need for empirical research to find out the solutions to free the hilsa fishers from the debt cycle created by the local money lenders.

Introduction of community-based sustainable fishery-based ecotourism: There are abundant opportunities for recreational fishing in Bangladesh that can bring many benefits to the country and can function as an essential tool for sustainable human development, including poverty alleviation, employment generation, and the development of rural areas [76]. Therefore, one option as part of fishery-based tourism could be in the form of fishing and dining cruises for locals as well as for international tourists. In such ventures, hilsa fishery stakeholders, including fishers, could engage in alternative income-generating activities.

Education and training programs for fishers: Diversification is necessary so that the fishers do not exclusively rely on one type of income, and for fishers to survive and increase their household income, more diverse strategies are needed other than relying solely on the catching sector [77]. The present study also found that alternative income source can reduce dependency on hilsa fishery and ensure food security for the fishers. However, proper education and training programs are needed for fishers to diversify their income. Government and local NGOs can take initiatives to train the fishers to start a business and to practice aquaculture, farming and gardening.

Social welfare and social security of the fishers: Fishing is a particularly hazardous occupation, with a relatively high rate of injury and death [78,79]. Hilsa fishing is not exceptional. Hence, hilsa fishers and their dependents need some form of protection in the event of injury, illness and death. In such cases, the Bangladesh government can introduce medical care, sickness benefits, unemployment benefits, old-age benefits, employment injury benefits, family benefits, maternity benefits, and invalidity benefits under the Fishermen’s Pension and Social Security Benefit Scheme. Such types of schemes are running in Srilanka [80]. However, it seems to be challenging to get fishers involved in any project requiring the regular payment of a premium because of the unpredictable nature of fishing incomes. Awareness programs through community organizations like fishery co-operatives can educate the fishers about such schemes.

Applying the concept of social-ecological traps: A socio-ecological trap is a concept in small scale fisheries referring to the dynamic interaction between poverty and natural resource use that creates situations considered undesirable in mainstream normative views of development [81,82]. A principal value of the social-ecological trap lens has been identified as its ability to highlight the interconnections between people and their natural environment, regarding them as elements of social-ecological systems [83]. The trap lens has been usefully applied both in and outside of small-scale fisheries settings in eastern Africa [81], Cameroon [84], Madagascar [83], in the lobster fishery industry in the eastern United States, and other settings [85]. The social-ecological trap lens has not been used to date in the small scale fishery settings of Bangladesh, including the hilsa fishery, despite its value in addressing how fishers’ experiences with poverty diminish their capacities to adapt to, or identify with, additional or alternative means or sources of livelihood. This is especially important in socio-ecological systems like the hilsa fishery as livelihood options, and, given weak or missing regulatory institutions, choices may well intensify local community dependence on the fishery and lead to overexploitation and eventual alterations of the fishery.

Co-management arrangements: Co-management can be a way of refining the social resilience of local communities, enabling them to have more power and control over decisions regarding how the natural resources they depend on are to be used [86]. Hence, the social resilience of the local hilsa fishery-based communities can be restored through co-management with the involvement of different stakeholders. However, the power realities under which hilsa fishers carry on their profession and the silvery catch finds its way to the consumer’s table within and outside the country needs further investigation. Additionally, research is necessary to find out critical power dynamics at play as to how administrative and regulatory oversight is exercised over the hilsa fishing sector.

A short summary of what responses/actions (according to their relative weight) can help to cope with what problems in hilsa fishery for sustainability is presented in Table 4. The ranking was based on the frequency and priority given to each by the informants in both interviews, FGDs and authors observations. Furthermore, this ranking was not undertaken quantitatively.

The framework that was employed in this paper—the DPSIR framework—provides an overall view on the relations of different aspects of SES, but it does not do such not within DPSIR categories. Along with DPSIR terminology, social and economic developments (driving forces, D) exert pressures (P) on the environment, and, consequently, the state (S) of the environment changes. This leads to impacts (I) on ecosystems, human health, and society, which may elicit a societal response (R) that feeds back on driving forces and direct pressures on state or impacts, via various mitigation efforts, adaptations, or remedial actions [34]. Furthermore, we have advanced the DPSIR framework by connecting specific solutions (responses) to specific problems (pressures) (Table 4). Additionally, we found there to be a general trade-off between providing an overall view of DPSIR on the one hand (Figure 3) and providing a more detailed view on the relationships between specific items within categories to each other. Since over ten plausible responses were identified, it could be constructive to assess which responses support each other and which are contradict each other. Future research could also assess possible synergies and incompatibilities between responses in the DPSIR framework depending on the surrounding societal trends and realities. However, we have opted to give a more general overview of DPSIR in the hilsa fishery, and we see this is important because the DPSIR framework has, to date (and to our knowledge), not been systematically applied to hilsa fisheries.

### 5.2. Limitation of the Study

The number of the respondents in this study was small, which may have created validity issues for this study; however, these issues are uncommon due to the study’s qualitative nature. Furthermore, the DPSIR framework in this study requires future work that should include follow-up work for developing the DPSIR framework as a decision support tool for fishery management. Additionally, elaborating the comparative severity of the presented impacts in the DPSIR framework would help strengthen the paper. This, however, remains a subject for future research.

## 6. Conclusions

The concepts of social-ecological systems (SES) and drivers-pressure-state-impact-responses (DPSIR) have been proven as feasible approaches to identify the challenges of small-scale fisheries to determine comprehensive social responses. Additionally, to ensure productive and sustainable fisheries, it is essential to understand the complex interactions between biology, environment, politics, management and governance. Worldwide fisheries are facing a range of challenges and lacking robust and careful management in place, levels of anthropogenic disturbance on ecosystems, and fisheries are likely to have a continuous negative impact on biodiversity and fish stocks. Fishery management authorities, therefore, need to be both efficient and effective in working towards long-term sustainable ecosystems and fisheries while also being resilient to political and socio-economic pressures.

The main goal of this study was to find out how to make hilsa production more sustainable and improve fishers’ socio-economic situation with the help of the DPSIR framework and by considering the hilsa fishery as an SES. To address these challenges, multi-level responses are recommended for the sustainability of the hilsa fishery, including the enhanced social resilience of the fishing community, increased incentives for all fishers and major stakeholders at the ground levels, promoted community-supported fisheries, improved financing mechanism for fishers, and the introduction of hilsa fish-based eco-tourism and alternative income-generating activities (AIGAs)—such as vegetable farming, livestock husbandry, aquaculture in the homestead ponds/cage culture in the canals/river as farming, and fisher women’s involvement in sewing and embroidery may be considered non-farming activities in the fishing communities to reduce pressure on hilsa fishing.

The hilsa fishery involves multiple stakeholders, and their participation is vital in implementing, monitoring, and enforcing regulations that can lead to compliance through collective action and can work well with institutions like intensives or compensation schemes. Additionally, the lack of proper policy implementation, participation of the local communities, and institutional collaborations have led to ineffective hilsa fishery management in Bangladesh. The timely enforcement of the policy, a close co-operation and mutual support among governmental institutions that have a similar scope and the local communities is therefore crucial for the sustainability of the hilsa fishery. It has been found that top-down government enforcement is not effective neither sustainable. Thus, a co-management approach could better provide both carrots and sticks for ensuring compliance with government rules and regulations.

To sum up, the present study summarizes and visualizes the cause-effect interactions between human pressures and environmental components in a manner familiar to fisheries managers and policymakers using the DPSIR framework, and it anticipates that this may help to bridge the gap between research and decision-making in the hilsa fishery.

## Figures and Tables

**Figure 1 ijerph-16-04814-f001:**
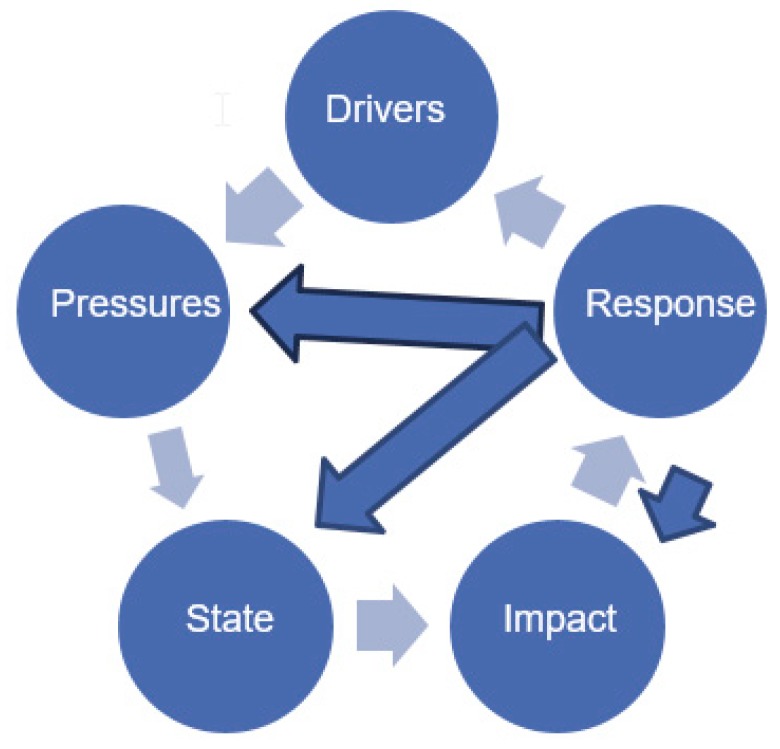
General drivers-pressure-state-impact-responses (DPSIR) framework [34].

**Figure 2 ijerph-16-04814-f002:**
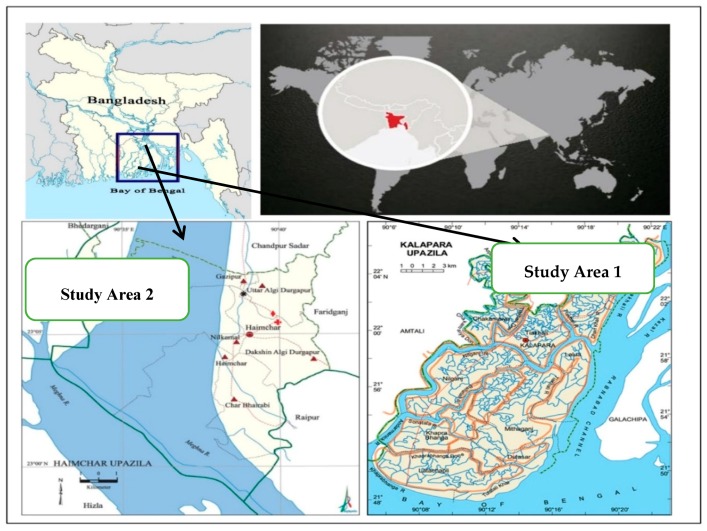
Location of the study villages (adopted from the Ministry of Local Government, Rural Development and Cooperatives, Bangladesh) [42,43].

**Figure 3 ijerph-16-04814-f003:**
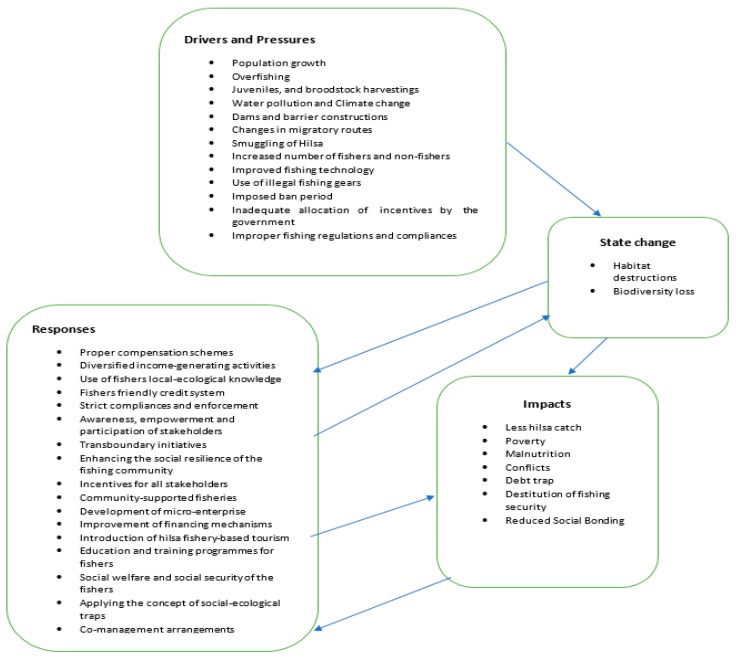
Proposal of the DPSIR framework for hilsa fishery management in Bangladesh.

**Table 1 ijerph-16-04814-t001:** Sample of interviewed hilsa fishery stakeholders.

Participants/Stakeholders	Number of Participants in Study Area 1		Number of Participants in Study Area 2	
	Rahmatpur	Sudirpur	Uttar Bagula	Dakxin Bagula
Hilsa fishers	Men = 15 Women = 5	Men = 15 Women = 5	Men = 15 Women = 5	Men = 15 Women = 5
Fish traders	2	2	2	2
Boat owners	2	2	2	2
Money Lenders/*Dadonder*	2	2	2	2
Local government representatives (Upazila Chairman, Union Parishad Chairman)	2	2	2	2
Local government administrative personnel	2	2	2	2

**Table 2 ijerph-16-04814-t002:** Representation in the combined stakeholders (n = 10) focus group discussions (FGDs) (one in each village).

Stakeholder Groups	Number of Participants
Hilsa fishers	4
Fish traders	1
Boat owners	1
Money Lenders/*Dadonder*	1
Local government representative (Upazila Chairman, Union Parishad Chairman)	1
Local governments personnel (Upazila Fishery Officer/Police)	1
Local NGOs representatives	1

**Table 3 ijerph-16-04814-t003:** Summary of drivers and pressures in the hilsa fishery.

Rank	Problem	Category	Effects	Solutions	Alternatives
1	Use of illegal fishing gear	Human	Less hilsa catching; loss of biodiversity	Enhanced compliance with regulations	Awareness, empowerment and participation; Alternative income-generating activities.
2	Improved fishing technology	Human	Less hilsa catching; loss of biodiversity	Enhanced compliance with laws	Awareness, empowerment and participation.
3	Population growth and increased number of fishers and non-fishers	Natural/Human	Less hilsa catching; poverty; malnutrition; conflicts and social tensions	Alternative income sources	Education
4	Overfishing		Less hilsa catching; poverty	Enhanced compliance with regulations and enforcements	Alternative income activities
5	Juveniles, and broodstock harvestings	Human	Less hilsa catching	Enhanced compliance with regulations and enforcements	Alternative income activities; Awareness, empowerment and participation
6	Imposed ban period		Malnutrition and the debt trap	Compensation-based schemes	Improved financing mechanisms; Alternative income activities; Other fish species catching allowed during the ban period
7	Inadequate allocation of incentives by the government	Government	Malnutrition, debt trap and social tensions	Incentives for the affected stakeholders	Compensation-based schemes
8	Improper fishing regulations and compliances	Government	Illegal fishing, debt trap	Enhanced compliance with laws and enforcements	Awareness, empowerment and participation
9	Dams and barrier constructions	Human	Siltation increased and migratory pattern of hilsa fish effected	Pre-planning and assessment before the establishment	Cooperation among neighboring countries including India and Myanmar
10	Water pollution and climate change	Human/Natural	Hilsa migration and hindrance for brood hilsa to lay eggs	Enhanced compliance with regulations and enforcements	Awareness, empowerment and participation
11	Changes in migratory routes	Natural/Human	Spawning grounds disturbed; reduced the nursery areas for the fish fry	Ecosystem-based management	Spatial closure in the mouth of the sanctuary
12	Smuggling of Hilsa	Human	Increasing hilsa selling price in the local market	Enhanced compliance with regulations and enforcements	Awareness, empowerment and participation

**Table 4 ijerph-16-04814-t004:** Summary of what responses/actions (according to their relative weight) can help to cope with what problems in hilsa fishery for sustainability.

Rank	Responses/Actions	Problems Addressed
1	Co-management arrangements	Enhance the social resilience of the stakeholders, power relations among stakeholders, and sustainable fishery management through participation
2	Enhance compliance with regulations/improved enforcement of the legislation	Sustainable hilsa fishery management and conservation of biodiversity in hilsa sanctuaries.
3	Incentives for all stakeholders	Social tensions/conflicts
4	Improved financing mechanism	Debt trap, poverty, alternative income-generating activities, and fishing ban period crisis
5	Compensation-based schemes	Incentives for fishers, conserve biodiversity, fishing ban period crisis, and alternative income-generating activities
6	Education	Creation of income sources and awareness
7	Social-ecological trap	Poverty, overexploitation of fishery resources, and alternative income-generating activities.
8	Awareness, empowerment and participation	Managing sanctuaries, biodiversity and conservation regulations, and monitoring and policing
9	Social resilience	Overexploitation of fishery resources, alternative income-generating activities, sharing responsibilities to manage fisheries, and community networks
10	Fishing-based tourism	Poverty and alternative income-generating activities
11	Promote local ecological knowledge (LEK)	Sustainable fishery management and selection of a sanctuary area
12	Transboundary initiatives	Sustainable fishery management, enforcement of ban period at the same time in Bangladesh, India, and Myanmar.
13	Social welfare	Protect fishers (injury, illness, death) and wellbeing of the fishers
14	Micro-enterprise	Fishing pressure, poverty, debt trap, helpful to buy fishing gears
15	Community-supported fisheries	Debt trap, to buy fishing equipment’s and to get a fairer market price for the fish catch

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
