# Peer review of "Understanding Social-Ecological Challenges of a Small-Scale Hilsa (Tenualosa ilisha) Fishery in Bangladesh"

_ijerph, 2019, doi:10.3390/ijerph16234814_

Round 1

Reviewer 1 Report

Dear Authors,

Thank you very much for this very interesting paper about the Hilsa fishery in Bangladesh. I have only a few minor comments.

Overall the description of the method, the research conducted and the results gives a very good overview of what are the problems of this fishery and what are possible ways to solve the overfishing situation etc. However, I think that it is not easy for the reader to see what are the main problems, which are of minor importance and what are especially solutions for the main problems (like overfishing or dependency of more and more people on the fishery for their subsistence). 

It would be helpful if you could reduce a bit the lengthy description of all the pressures and drivers to free space for some kind of summary: what are the main problems which a management need to address? Are there alternatives for the fishers in case an improved management will be implemented? Etc.

This would mean that you can also concentrate in the part on the solutions on the main issues. You propose a lot of actions which could solve some of the problems. For the reader, however, it is not clear which of the actions have a higher priority than others. To give an example: would introduction of some kind of community based or co-management be helpful to integrate the fishers in the management on one side and make the enforcement better on the other side. I assume that governmental enforcement (as mentioned briefly) showed to be not that effective - seeing this large numbers of fishers and the huge area to cover. 

I feel also a bit lost what can be done now. It is clear that there need to be changes to solve the problems in the fishery. However, the pressure on the system increased because more and more people need to be fed from the resource. Is there any chance for alternatives (other jobs, other sources of food, etc.)? Otherwise there will be no strict management as the people need to eat something. 

Author Response

Response to Reviewer 1

We thank you for having taken the time to review our article and for providing some very useful and valuable comments which we believe have helped to further improve the article. Below we provide our responses to your query-

Point 1: Overall the description of the method, the research conducted, and the results gives a very good overview of what are the problems of this fishery and what are possible ways to solve the overfishing situation etc. However, I think that it is not easy for the reader to see what the main problems are, and what are especially solutions for the main problems (like overfishing or dependency of more and more people on the fishery for their subsistence).

Response 1: We have considered this point and have incorporated as follows. At the beginning of the Results section (lines 224-234), we had a paragraph (copy-pasted here below) stating the main problems of this fishery, as well as the solutions proposed for these problems:

“The results suggest that the main driving forces for less catch by fishers are: use of illegal fishing gears, overpopulation in the coastal areas, overfishing, harvesting of juveniles, river water pollution, climate change, dam constructions in the upstream, and cross-border smuggling of hilsa. These challenges have led to both compromise with management strategies as well as disincentives, thereby further increasing the pressures on the hilsa fishery. These pressures include habitat destruction and biodiversity loss, which in turn result in less hilsa catch, poverty, malnutrition, stakeholder conflicts, insecurity and social tensions. To address these challenges, multi-level responses are recommended for the sustainability of the hilsa fishery, including enhanced social resilience of the fishing community, increased incentives for all fishers and major stakeholders at the local level, promotion and support for community-managed fisheries, improved financing mechanisms for the fishers, and other sustainable economic activities such as small-scale community-managed eco-tourism.”

Moreover, in the revised manuscript, a short summary of the results has now been added, including a new table (see Table 3) outlining the main findings, as advised by the reviewers. Also, we now have in the Discussion section a summary of the responses (lines 727-731) that we identified as possible solutions to address the problems in the hilsa fishery.

Point 2: It would be helpful if you could reduce a bit the lengthy description of all the pressures and drivers to free space for some kind of summary: what are the main problems which a management need to address? Are there alternatives for the fishers in case an improved management will be implemented?  Etc.

Response 2: We fully agree with this comment, and to a large extent we have addressed this point already in the revisions we made in the manuscript in response to Point 1 (see above). In addition, we have now revised the drivers and pressures section of the Results, reducing it in length and including a summary of the main problems and management strategies that need to be addressed (see Table 3). The summary we have added in the revised manuscript is drawn from the conducted interviews and Focus Group Discussions (FGDs) with the hilsa fishery stakeholders (please see the supplementary file – Fishing and fish behaviour). In other words, we summarized the findings as approved by the fishers and other stakeholders during the FGDs and ranked these according to the importance given by our interviewees and FGD participants.

Point 3: This would mean that you can also concentrate in the part on the solutions on the main issues. You propose a lot of actions which could solve some of the problems. For the reader, however, it is not clear which of the actions have a higher priority than others. To give an example: would introduction of some kind of community based or co-management be helpful to integrate the fishers in the management on one side and make the enforcement better on the other side. I assume that governmental enforcement (as mentioned briefly) showed to be not that effective - seeing this large numbers of fishers and the huge area to cover.

Response 3:  Once again, we agree with this comment and have incorporated the suggestions into the revised manuscript. In the revised manuscript Discussion section Table 4 we placed the actions or responses according to their priority. Such interpretation was based on respondent’s priority and authors observations.

 Point 4: I feel also a bit lost what can be done now. It is clear that there need to be changes to solve the problems in the fishery. However, the pressure on the system increased because more and more people need to be fed from the resource. Is there any chance for alternatives (other jobs, other sources of food, etc.)? Otherwise there will be no strict management as the people need to eat something.

 Response 4: Yes, these are points of concern that indeed need to be addressed more clearly. We believe that with the newly incorporated tables (Table 3 and Table 4), we now exemplify more clearly what needs to be done to deal with the increasing pressures on the socio-ecological system of the hilsa fishery. Our main recommendations are also summarized in the conclusion as follows:

The main goal of this study was to examine whether and how hilsa production could be rendered more sustainable, also improving fishers’ socio-economic situation. We undertook the study with the help of the DPSIR framework and considering the hilsa fishery as an SES. The hilsa fishery involves multiple stakeholders, and their active participation is vital in all phases of identifying, planning, implementing, monitoring and enforcing regulations that can lead to better compliance through collective action. Also, the lack of proper policy implementation, as well as the limited participation of local communities and institutional collaborations, all result in ineffective hilsa fisheries management in Bangladesh. Timely enforcement of the policy, supported by a close co-operation and mutual support between governmental institutions and local communities, is crucial for the long-term sustainability of the hilsa fishery.”

Alternative income-generating activities (AIGAs) including the vegetable farming, livestock husbandry, aquaculture in the homestead ponds/cage culture in the canals/river as farming; and fishers’ women involvement in sewing and embroidery may be considered non-farming activities in the fishing communities to reduce pressure on hilsa fishing.

Reviewer 2 Report

Bangladesh has ranked third in the world in terms of inland fish production in 2018, according to a report by the Food and Agriculture Organization (FAO). Thus, inland fisheries are very important in this country. This long manuscript deals with problems and reasons in this activity by using interview and discussions with stakeholders.

I however think this manuscript is sometimes quite emotional and therefore this should not be published unless following questions are reasonably justified in the manuscript.

There is no evidence of declined catch of hilsa in Bangladesh. According to the fisheries statistics in FAO, annual hilsa production in Bangladesh is continuously increasing since 2003 so that it is hard to believe the declined catch. Authors should present scientific data/statistics to prove the hilsa fishery in those study areas have encountered such problems. Also should justify that study areas can be typical examples of hilsa fishery/fishing communities in the country/region.

I found such kind of prejudices in some parts of the text. For instance, authors showed the “Population growth” is a one of Drivers, but it is also not supported by data/statistics, so we can not understand how much population increased and its impact.

I do not see any reasons on why authors selected stakeholders of different characteristics (fishers, traders, administrators, etc.), because there were no different opinions to the specific topics in the text. For instance, one fisher stated his opinion about the allocation of incentives, but just that. I wonder if authors interviewed same topic to an administrator, not a fisher, he/she would say different opinion. Thus this manuscript seems to be one-sided. Biological/ecological/technical characteristics of hilsa fishery are not well-explained at all so that readers will not be able to understand where the problem is.

best,

Reviewer 3 Report

TSG’s review of IJERPH-620825 paper on SSF hilsa fishery in Bangladesh 17.10.19

This paper is a study of four small-scale hilsa chad fisheries in Bangladesh which are not in good condition. The authors examine the causes of the problems these fisheries face, the socio-ecological impacts of these problems, and the fishers’ views of what the government’s response should be to these impacts, before setting out their own recommendations for the improvement of the fisheries.

The paper is clear, well written, and coherently organised, providing a comprehensive and graphic picture of the dire situation experienced by these small-scale fishers (SSF). The first author carried out 120 key informant interviews and eight focus group discussions during 2016-2019 to obtain primary data, and conducted a literature search and newspaper review to obtain secondary data. In my view, the paper is publishable because it provides new material on an important fishery, revealing a familiar pattern of pressure on SSF in developing countries.

My only reservation is that the authors might have been more interpretive in the way they present their findings. For example, in section 4.1 - Drivers and Pressures - the authors list 13 causes of the problems, yet they never indicate a rank order of the relative seriousness of these 13 causes. I would like them to have told us which are the most important causes; which are the most intractable causes; which causes arose first; and what are the relationships between the different causes. Also, the authors could have grouped the causes into categories such as natural, human, governmental. In section 4.3 – Impacts - the authors list seven impacts, but never compare their respective significance. In section 4.4 – Responses – the authors list seven responses urged by fishers, but again do not tell us how the fishers ranked them in importance. In section 5.1 – Elaborating policy responses to enhance sustainability in and around hilsa fisheries – the authors list 13 recommendations of their own, but once more without any indication of their relative weight. I think the paper could be improved if the authors explained how the items within each of these four lists related to each other. As they stand, the items in them seem randomly listed - plucked out of the air like items in laundry lists.

Line 701: what is a ‘socio-ecological trap’?
